# Genetic transformation of the dinoflagellate chloroplast

Isabel C Nimmo[1], Adrian C Barbrook[1], Imen Lassadi[1], Jit Ern Chen[2,3], Katrin Geisler[4], Alison G Smith[4], Manuel Aranda[2], Saul Purton[5], Ross F Waller[1], R Ellen R Nisbet[1]*, Christopher J Howe[1]

[1]Department of Biochemistry, University of Cambridge, Cambridge, United Kingdom; [2]Red Sea Research Center, King Abdullah University of Science and Technology (KAUST), Thuwal, Saudi Arabia; [3]Jeffrey Sachs Center on Sustainable Development, Sunway University, Bandar Sunway, Malaysia; [4]Department of Plant Sciences, University of Cambridge, Cambridge, United Kingdom; [5]Institute of Structural and Molecular Biology, University College London, London, United Kingdom

**Abstract** Coral reefs are some of the most important and ecologically diverse marine environments. At the base of the reef ecosystem are dinoflagellate algae, which live symbiotically within coral cells. Efforts to understand the relationship between alga and coral have been greatly hampered by the lack of an appropriate dinoflagellate genetic transformation technology. By making use of the plasmid-like fragmented chloroplast genome, we have introduced novel genetic material into the dinoflagellate chloroplast genome. We have shown that the introduced genes are expressed and confer the expected phenotypes. Genetically modified cultures have been grown for 1 year with subculturing, maintaining the introduced genes and phenotypes. This indicates that cells continue to divide after transformation and that the transformation is stable. This is the first report of stable chloroplast transformation in dinoflagellate algae.
DOI: https://doi.org/10.7554/eLife.45292.001

*For correspondence:
rern2@cam.ac.uk

**Competing interests:** The authors declare that no competing interests exist.

## Introduction

Coral reefs are complex ecosystems, made up of many thousands of species. At the base of the ecosystem are dinoflagellate algae, frequently referred to as zooxanthellae. These single-celled algae live in symbiosis with corals as intracellular photosynthetic symbionts, providing fixed carbon to the host. Loss of the symbiotic alga results in coral bleaching, which is one of the most urgent and worsening worldwide ecological concerns. In 2016, 85% of the Great Barrier Reef was found to be affected by coral bleaching, a significantly higher proportion than had been previously identified (*Hughes et al., 2017*).

Change in sea water temperature is recognized as one of the environmental causes of coral bleaching (*Spalding and Brown, 2015*). It is likely that this results in disturbance of photosynthetic electron transfer in the dinoflagellate symbiont and consequent damage (*Rehman et al., 2016*), (*Slavov et al., 2016*). The PsbA (D1) reaction center protein of photosystem II is believed to be an important target of such damage (*Warner et al., 1999*). The key subunits of photosynthetic electron transfer chain complexes, including the PsbA protein, are encoded in the dinoflagellate chloroplast genome (*Howe et al., 2008*). There have been no reports to date of transformation of the dinoflagellate chloroplast genome, hampering attempts to study the mechanism of bleaching.

An alternative approach to transformation of the chloroplast might be to insert genes for proteins carrying chloroplast targeting sequences into the nucleus. There have been numerous attempts at stable nuclear transformation of dinoflagellates, but none has been clearly successful. An early report

of the transformation of the dinoflagellates *Amphidinium sp.* and *Symbiodinium microadriaticum* mediated by silicon carbide whiskers, with selection for resistance to hygromycin or G418 and using β-glucuronidase (GUS) as a reporter, appeared to produce transformants after 12 weeks (*ten Lohuis and Miller, 1998*). However, there are no reports of successful use of this technique since the initial publication. In 2019, a series of experiments on *Symbiodinium microadriaticum,* using biolistics, electroporation and agitation with silicon carbide whiskers failed to introduce a chloramphenicol acetyl transferase (CAT) gene to the nuclear genome (*Chen et al., 2019*).

Despite this, Ortiz-Matamoros and co-workers reported transient expression of GFP in *Symbiodinium* using a plasmid designed for plant transformation introduced by treatment with glass beads and polyethylene glycol, and selection for resistance to the herbicide Basta (gluphosinate) (*Ortiz-Matamoros et al., 2015a*). However, transformed cells were not capable of cell division, and no genetic confirmation of transformation was carried out. Transformation with the same plasmid mobilized by the plant pathogen *Agrobacterium* was also reported, although the transformed cells again failed to divide (*Ortiz-Matamoros et al., 2015b*). The lack of stable expression of heterologous genes limits the use of these techniques for functional biochemical studies.

Here, we describe a method for stable transformation of the dinoflagellate chloroplast. The chloroplast genome of dinoflagellate species containing the carotenoid peridinin (which is the largest group and includes those forming symbionts with coral) is typically fragmented, comprising approximately 20 plasmid-like DNA molecules of 2–5 kbp known as 'minicircles' (*Zhang et al., 1999*), (*Barbrook et al., 2014*). Each minicircle typically carries a single gene, together with a conserved core region which is assumed to contain the origin of replication as well as the transcriptional start site (*Howe et al., 2008*). These minicircles have been shown to be localized to the chloroplast using in situ hybridization (*Owari et al., 2014*). Each chloroplast contains multiple copies of each minicircle, although the exact copy number varies according to the growth stage of a culture (*Koumandou and Howe, 2007*).

We exploited this unusual minicircular genome organization to create shuttle vectors for dinoflagellate chloroplast transformation. We created two artificial minicircles, both based on the *psbA* minicircle from the dinoflagellate *Amphidinium carterae,* as the *A. carterae* chloroplast genome is the best characterized amongst dinoflagellates (*Barbrook et al., 2012*; *Koumandou et al., 2004*; *Barbrook et al., 2001*). We replaced the *psbA* gene with a selectable marker (either a modified version of *psbA* which confers tolerance to the herbicide atrazine (*Hirschberg and McIntosh, 1983*), or a gene for chloramphenicol acetyl transferase (CAT), which confers resistance to chloramphenicol), and an *E. coli* plasmid backbone (to allow propagation in *E. coli*). We tested numerous transformation methods and were able to obtain sucessful introduction of these artificial minicircles into dinoflagellates using particle bombardment. Following selection, we could detect the presence of the artificial minicircles, and transcripts from them, using PCR and RT-PCR. We were able to detect the product of the introduced chloramphenicol acetyl transferase gene using immunofluorescence microscopy. Cultures under selection continued to divide and maintain the artificial minicircles for at least 1 year, indicating that transformation was stable. The availability of a method for dinoflagellate chloroplast transformation enables a range of studies on the maintenance and expression of this remarkable genome and the proteins it encodes, such as PsbA.

## Results

### Construction of artificial minicircles

Two artificial minicircles were used in this study. The first, pAmpPSBA, was designed to confer atrazine tolerance. Tolerance to atrazine in plants can be conferred by a single residue change in the PsbA protein, where a Serine is mutated to a Glycine (*Goloubinoff et al., 1984*). We therefore cloned the *A. carterae psbA* minicircle into the *E. coli* vector pGEM-T easy (Promega) and introduced the necessary mutations into the *psbA* gene using site-directed mutagenesis, *Figure 1*. The second artificial minicircle, pAmpChl, was designed to confer chloramphenicol resistance. It is also based on the *A. carterae psbA* minicircle, but the *psbA* gene was excised and replaced by a *A. carterae* codon-optimized gene encoding chloramphenicol acetyl transferase. The plasmid backbone is *E. coli* pMA, *Figure 1*.

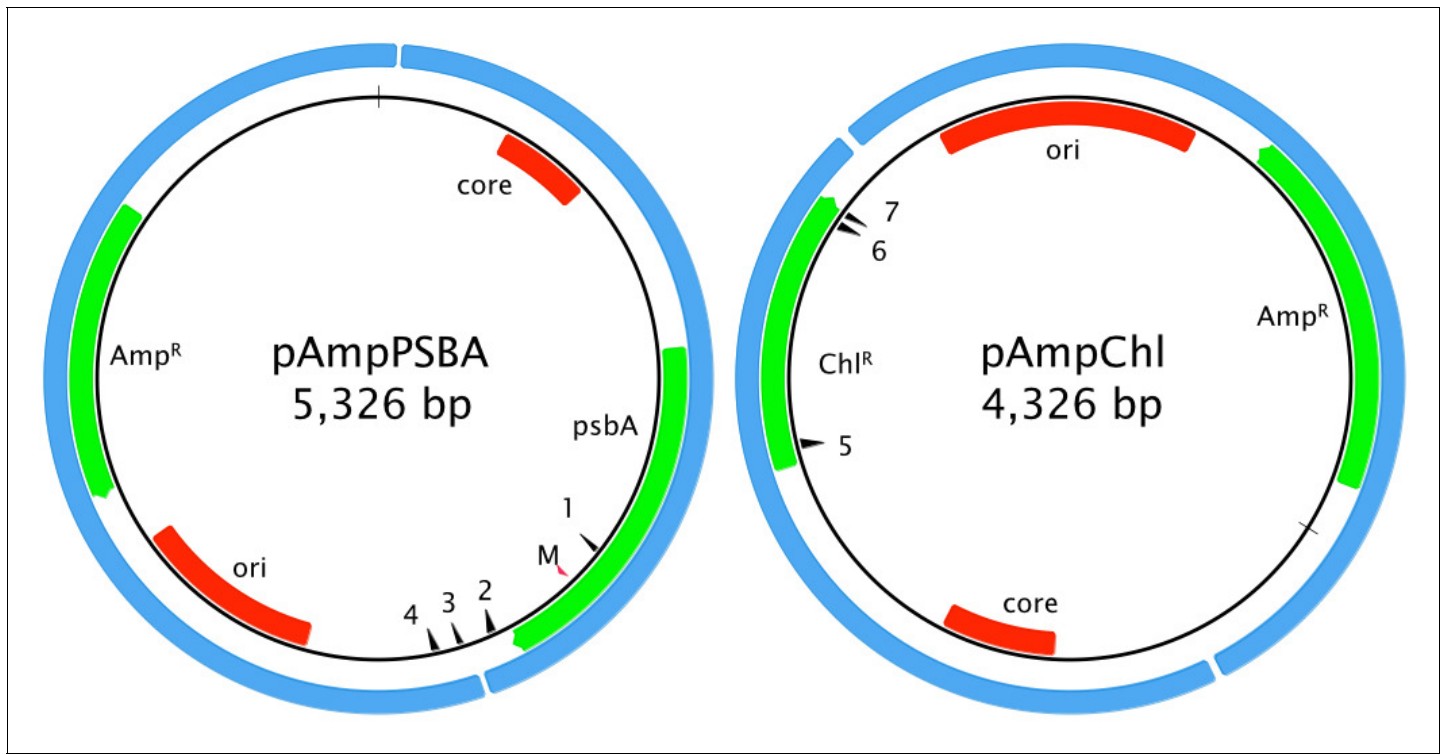

**Figure 1.** Artificial minicircle design. Left, pAmpPSBA. Right, pAmpChl. Origins of replication (*ori* for *E. coli* plasmid, core region for *A. carterae* minicircle) are shown in red. Protein-coding genes (encoding ampicillin resistance, chloramphenicol resistance or PsbA) are shown in green. The blue shows the original source of of the genetic material (*E. coli* plasmid and *A. carterae* minicircle). The red arrow showing the position of the mutation in *psbA* that confers resistance to atrazine is marked with 'M'. Primer sites pAmpPSBA 1: MC-pG-F-II, 2: MC-pG-F, 3: MC-pG-R-II, 4: MC-pG-R; pAmpChl: 5: CAT-R, 6: CAT-F, 7: CAT-FSS. CAT-F-Nest and CAT-R-Nest are immediately adjacent to CAT-F and CAT-R, respectively.
DOI: https://doi.org/10.7554/eLife.45292.002

## Glass beads and electroporation

There has been one previous report of stable transformation of dinoflagellates, using silicon carbide whiskers (*ten Lohuis and Miller, 1998*). This has never been reproduced, despite numerous attempts (*Walker et al., 2005*). Additionally, the whiskers are a significant health hazard, and, in other species, glass bead-mediated transformation has a higher transformation efficiency (*Ortiz-Matamoros et al., 2018*). There has been one previous report of transient transformation of dinoflagellates with glass beads (*Ortiz-Matamoros et al., 2015a*). We therefore first sought to replicate this finding, by transforming the *A. carterae* dinoflagellate chloroplast genome using glass beads. We used the artificial minicircle pAmpPSBA followed by selection with atrazine. The experiment was carried out three times, but no viable cells were seen following selection (i.e. following the addition of atrazine, all cells died at the same time that untransformed cells died).

Polyethylene glycol has been reported to increase glass bead transformation efficiency. We therefore added polyethylene glycol and repeated the transformation (in triplicate). No live cells were recovered following selection, indicating that this method did not give rise to stable transformants. A second attempt (in triplicate) was also not successful.

Many eukaryotic protist species can be transformed using electroporation, and the Lonza Nucleofector system is used with many hard-to-transfect species. This includes *Perkinsus marinus*, a sister group to the dinoflagellates (*Burkett and Vasta, 1997*). The artificial minicircle pAmpPSBA was used to attempt to transfect *A. carterae* using the Lonza Nucleofector with several settings indicated as suitable for protist transformation (X-100, D-023, L-029 and EH 100), each in triplicate. We were never able to recover any transformants.

## Biolistic transformation with pAmpPSBA

We next turned to biolistics, since this has been reliably used to transform the unicellular alga *Chlamydomonas* for 30 years (*Boynton et al., 1988*). Nine experiments to introduce the pAmpPSBA artificial minicircle into *A. carterae* using biolistic transformation were carried out, using a range of rupture disk pressures. Each experiment was carried out in triplicate, and included a single negative control line (cells subjected to biolistic treatment but without pAmpPSBA). The mean survival time for each culture under selection was assessed by bright field microscopy, and results are shown in *Table 1*. In six experiments, *A. carterae* cells shot with particles carrying pAmpPSBA showed greater mean survival time under selection conditions than untransformed cells, suggesting successful transformation (Experiments A2, A3, A4, A5, A7 and A8). One experiment (A6) was harvested prior to the death of the control strain, so no conclusions can be drawn on relative survival times. Finally, two experiments showed no difference in the length of time for which cells survived. The first (Experiment A1) was carried out using the lowest pressure rupture disks. In this experiment, control and experimental cells survived just 13 days, suggesting that cells had not been transformed, perhaps because an insufficient bombardment velocity had been applied. In the second experiment (A9), selection was carried out with 1 $\mu$g ml$^{-1}$ atrazine, below the lethal concentration of 2 $\mu$g ml$^{-1}$. Both test and control cultures survived at least three months, with subculturing occurring at 8-week intervals.

## Biolistic transformation with pAmpChl

Transformation attempts were also made with *A. carterae* using chloramphenicol resistance as selectable marker. Experiments were carried out with pAmpChl and 1550 p.s.i. rupture disks. In the first experiment, chloramphenicol (final concentration 10–50 $\mu$g ml$^{-1}$) was applied after 3 days in liquid culture, to allow time for initial synthesis of chloramphenicol acetyl transferase (Experiments C1A-E), *Table 2*. No untransformed wild-type cells (i.e. shot with gold particles without DNA) survived after 15 days, whatever the chloramphenicol concentration. However, at 10 $\mu$g ml$^{-1}$ chloramphenicol, cells shot with particles carrying the pAmpChl plasmid survived for at least 35 days, Experiment C1A. When chloramphenicol concentration was 30 $\mu$g ml$^{-1}$ or greater, cells shot with particles containing the pAmpChl plasmid had died by day 15, (Experiment C1C-E), *Table 2*. Note that where appropriate, cells were subcultured after 28 days.

## Detection of artificial minicircles using PCR

To test if the transformation construct could be recovered from putatively transformed cultures, DNA was isolated from atrazine-selected *A. carterae* cultures (experiment A5, two lines designated A5.1 and A5.2) by vortexing with glass beads. DNA was also isolated from wild-type *A. carterae* as a

**Table 1.** Biolistic transformation of *A. carterae* with pAmpPSBA.

Each experiment was carried out in triplicate, thus producing three potentially transformed lines. In addition, one line of cells was subjected to biolistic bombardment, but without the pAmpPSBA ('untransformed'). Note that cultures from experiments 5–9 were harvested for genetic analysis, and thus the listed survival time is the day of harvesting, labeled with *.

| Experiment | Rupture disk (p.s.i.) | Atrazine concentration ($\mu$g ml$^{-1}$) | Survival untransformed (days) | Mean survival pAmpPSBA (days) |
|---|---|---|---|---|
| A1 | 1100 | 2.5 | 13 | 13 |
| A2 | 1350 | 2.5 | 13 | 16 |
| A3 | 1350 | 2.5 | 13 | 15 |
| A4 | 1550 | 2.5 | 13 | 17 |
| A5 | 1550 | 2.5 | 12 | 20* |
| A6 | 1550 | 2 | 7* | 7* |
| A7 | 1550 | 2 | 13 | 13* |
| A8 | 1550 | 2 | 12 | 15* |
| A9 | 1550 | 1 | 3 months* | 3 months* |

DOI: https://doi.org/10.7554/eLife.45292.003

**Table 2.** Biolistic transformation of *A. carterae* with pAmpChl.

Each experiment was carried out in triplicate, thus producing three potentially transformed lines. In addition, one line of cells was subjected to biolistic bombardment, but with gold particles lacking the pAmpChl ('untransformed'). For experiment 1, cells from each plate (three shot with gold particles carrying the plasmid and one with gold particles only) were divided into five separate samples, each incubated at a different chloramphenicol concentration. Note that cultures from experiments C1A, C2, and C3, were harvested for genetic analysis, and thus the listed survival time of lines still alive at that point is the day of harvesting, labeled with *. Experiment C4 was still alive at 57 weeks and is thus marked with +.

| Experiment | Rupture disk (p.s.i.) | Chloramphenicol concentration ($\mu g\ ml^{-1}$) | Survival untransformed (days) | Mean survival pAmpChl (days) |
|---|---|---|---|---|
| C1A | 1550 | 10 | 15 | 35* |
| C1B | 1550 | 20 | 15 | 17 |
| C1C | 1550 | 30 | 15 | 15 |
| C1D | 1550 | 40 | 15 | 15 |
| C1E | 1550 | 50 | 13 | 15 |
| C2 | 1550 | 10 | 13 | 13* |
| C3 | 1550 | 10 | 14* | 14* |
| C4 | 1550 | 20 | 16 | 57 weeks + |

DOI: https://doi.org/10.7554/eLife.45292.004

negative control. In addition, a DNA purification was carried out with transformed cells (line A5.1), but without vortexing with glass beads ('unbroken cells') in order to test whether DNA remained adsorbed to the outside of cells. A positive control was included using the pAmpPSBA artificial minicircle. PCR was performed using the primers MC-pG-F and MC-pG-R (*Figure 1*) which lie on either side of the junction between the *psbA* minicircle and the pGEM-T Easy vector. A single product was amplified from each of lines A5.1 and A5.2, with no product detected from either the wild type or the 'unbroken cells' (*Figure 2A*). This product matched the size of product from the positive control. Products from both lines were cloned and sequenced. The sequence was as expected from pAmpPSBA, as a chimaera between the *psbA* minicircle and the pGEM-T Easy vector, confirming that the atrazine-resistant *A. carterae* did indeed contain the pAmpPSBA sequence.

To test if the pAmpChl artificial minicircle was present, DNA was isolated from three chloramphenicol-selected *A. carterae* lines (C1A.1, C1A.2 and C1A.3) after 35 days of selection. PCR using primers within the chloramphenicol resistance gene (CAT-F and CAT-R) was carried out. A positive control was included using the pAmpChl artificial minicircle. A single product was obtained from lines C1A.1 and C1A.3, with no product detected from line C1A.2 (*Figure 2B*). Products from lines C1A.1 and C1A.3 were cloned and sequenced. The sequence was the same as that expected from pAmpChl, which confirmed that the chloramphenicol-resistant *A. carterae* did indeed contain the pAmpChl sequence.

## Relative copy number of artificial minicircles

As we utilized the backbone of an existing *psbA* minicircle to create pAmpChl we tested to see what proportion of the *psbA* minicircles contained the *psbA* gene and what proportion contained the CAT gene (and associated shuttle vector). We therefore designed primers immediately flanking the *psbA* gene (copy-F and copy-R), in a region that was common to both the *psbA* and pAmpChl minicircles, and carried out PCR. This should amplify either *psbA* or CAT. Analysis of the products using agarose gel electrophoresis revealed the presence of two bands, one corresponding to the *psbA* gene and one to the CAT gene, as shown in *Figure 3*. Taking into account the size of the products, the likelihood that the CAT-specific product would be generated more efficiently, being smaller, and the relative intensity of each band, it would appear that there are roughly similar numbers of each minicircle in the chloroplast.

## Transcription of the artificial minicircles

In order to test if transcripts from the two artificial minicircles could be detected in the putatively transformed lines, total RNA was extracted and purified. cDNA was synthesized using RNA from

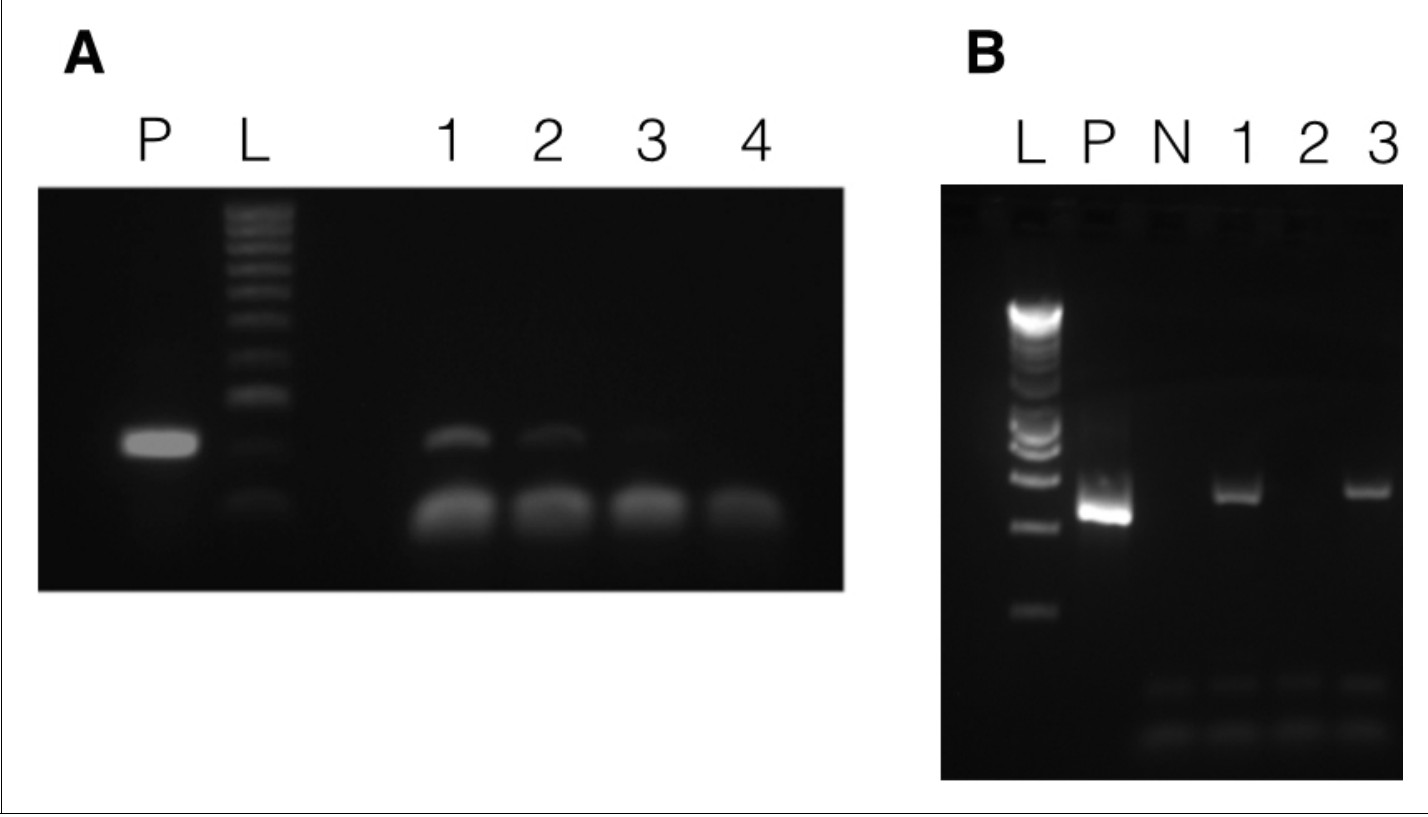

**Figure 2.** Presence of vectors in transformed *A. carterae*. Panel A shows results with pAmpPSBA. DNA was isolated from cells putatively transformed with pAmpPSBA, and a PCR reaction performed to amplify a ~ 200 bp region in the plasmid. Lane P, positive control (PCR with plasmid only), Lane L, Hyperladder 100 bp (Bioline) marker, Lane 1, transformed cell line A5.1 (strong band), Lane 2, transformed cell line A5.2 (faint band), Lane 3, transformed cell line A5.1 but without cells being broken open prior to DNA extraction (no band), Lane 4, wild type (i.e. untransformed cells, no band)). Panel B shows results with pAmpChl. DNA was isolated from cells putatively transformed with pAmpChl, and a PCR reaction performed to amplify a 580 bp region in the plasmid. Lane L, Hyperladder 1 kb (Bioline) marker, Lane P, positive control (PCR with plasmid only), Lane N, negative control (no template), Lane 1, transformed cell line C1A.1, Lane 2, transformed cell line C1A.2, Lane 3, transformed cell line C1A.3. Apparent differences in mobility between bands in lanes P, 1 and 3 are due to gel loading.

DOI: https://doi.org/10.7554/eLife.45292.005

atrazine-selected cultures from experiment A6 (lines A6.1, A6.2, A6.3 and untransformed) and random hexamer primers, followed by PCR with the specific primers MC-pG-F-II and MC-pG-R-II. cDNA was synthesized using RNA from chloramphenicol-selected culture lines C3.1, C3.2 and C3.3 and the gene-specific primer CAT-FSS, followed by a nested PCR strategy. Primers CAT-F and CAT-R were used in the first round of PCR (30 cycles). 1 µl of PCR product was used as template for the second round of PCR (10 cycles) with primers CAT-F-Nest and CAT-R-Nest. Negative controls, which omitted the reverse transcriptase, were included for all RT-PCRs. A positive control was included using the pAmpPSBA or pAmpChl artificial minicircle.

RT-PCRs using RNA from the three atrazine-selected lines in Experiment A6 all yielded a band consistent with the size of the positive control, *Figure 4A*. The DNA in the bands was purified, cloned and sequenced. The sequence of all three matched the pAmpPSBA artificial minicircle, confirming that it was transcribed. The sequence spanned the site of the atrazine resistance mutations and included the expected sequence alterations. The negative control yielded no PCR products (data not shown). The same results were obtained for three lines in each of Experiments A7 and A8 (data not shown).

RT-PCRs using RNA from the three chloramphenicol-selected lines (C3.1-C3.3) yielded bands from two of the three cell lines in Experiment C3 (*Figure 4B*). The PCR products were sequenced directly and shown to correspond to the pAmpChl minicircle. The negative control yielded no PCR product. Two of three lines in Experiment C2 yielded bands in RT-PCRs (data not shown).

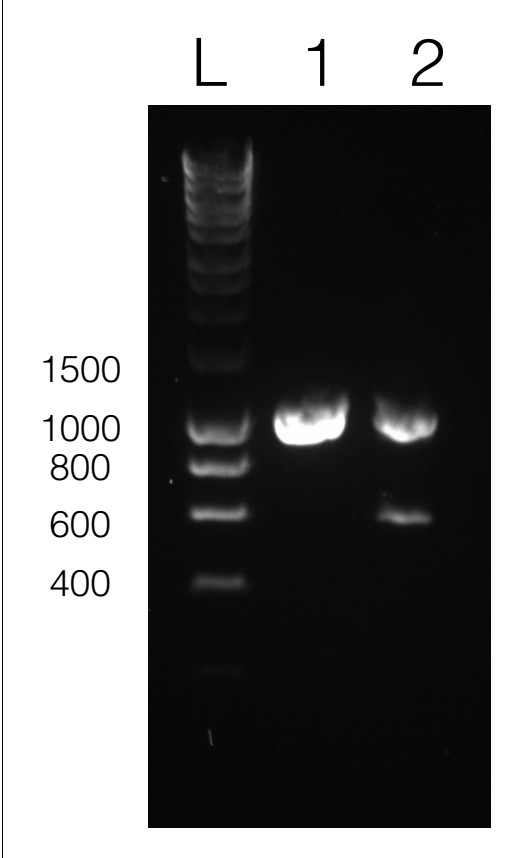

**Figure 3.** Relative copy number of *psbA* and pAmpChl minicircles. DNA was isolated from cells transformed with pAmpChl, and a PCR reaction performed to amplify either psbA or CAT. Lane L, Hyperladder 1 kb (Bioline) marker, Lane 1, wildtype cells (amplifying *psbA* only), Lane 2 pAmpChl line (amplifying *psbA* (top band) and CAT (lower band)).

DOI: https://doi.org/10.7554/eLife.45292.006

## Artificial minicircle products localize to the chloroplast

In order to confirm that we had transformed the chloroplast genome and not the nuclear genome, we carried out an immunofluorescence assay with an anti-chloramphenicol acetyl transferase antibody, together with a secondary antibody labeled with Alexa Fluor 405. The antibodies co-localized with the chloroplast, identified as the region of the cell with significant autofluorescence, as shown in *Figure 5*. The antibodies did not localize with the nucleus or other parts of the cell. This result shows that the CAT protein is expressed in the chloroplast (compare the merged image with the chloroplast autofluorescence and CAT images). As the CAT gene did not include a chloroplast targeting sequence (which would be necessary to target nuclear-encoded proteins to the chloroplast), the pAmpChl artificial minicircle must be located in the chloroplast.

## Stability of transformation

To test if the atrazine-resistance phenotype transformation of the dinoflagellate chloroplast was stable under low-level selection, cells were shot with gold particles carrying pAmpPSBA and cultured under continuous atrazine selection at 1 µg ml$^{-1}$ (experiment A9 in *Table 1*). Cell counts increased over time, although at a rate much lower (~10%) than untransformed cells under no selection. An untransformed cell line was also maintained, which survived under the same atrazine concentration (1 µg ml$^{-1}$). Both cell lines were subcultured at 8-week intervals. After 3 months, cells were harvested and DNA was isolated. PCR using the primers MC-pG-F and MC-pG-R was carried out (as above). A positive control PCR was included using the pAmpPSBA vector, and PCR with DNA isolated from the untransformed, wild type cells maintained at non-lethal atrazine concentration was included as a negative control (*Figure 6A*). A single product, of expected size, was obtained using the three transformed cell lines, with no product detected from the untransformed cells. DNA sequencing confirmed that the products from all three lines corresponded to pAmpPSBA. This showed that the transformation of *A. carterae* with pAmpPSBA was stable at a non-lethal atrazine concentration.

To test if the chloramphenicol-resistance phenotype transformation of the dinoflagellate chloroplast was stable, three lines were generated by shooting with gold particles carrying pAmpChl and cultured under continuous chloramphenicol selection at 20 µg ml$^{-1}$ (Experiment C4 in *Table 2*) with subculturing every 14 days. After 18 weeks, well after control untransformed cells had died, a sample of cells was harvested from each transformed line and DNA was isolated. A PCR reaction using the primers CAT-F-Nest and CAT-R-Nest was carried out on each sample. A positive control PCR was included using the pAmpChl vector (*Figure 6B*). A single product, of expected size, was obtained for each of the three transformed cell lines. DNA sequencing confirmed that the products from all three lines corresponded to pAmpChl. No band was seen in a wild-type PCR carried out with the same primers on wild-type cells (data not shown), confirming that the band could only have arisen

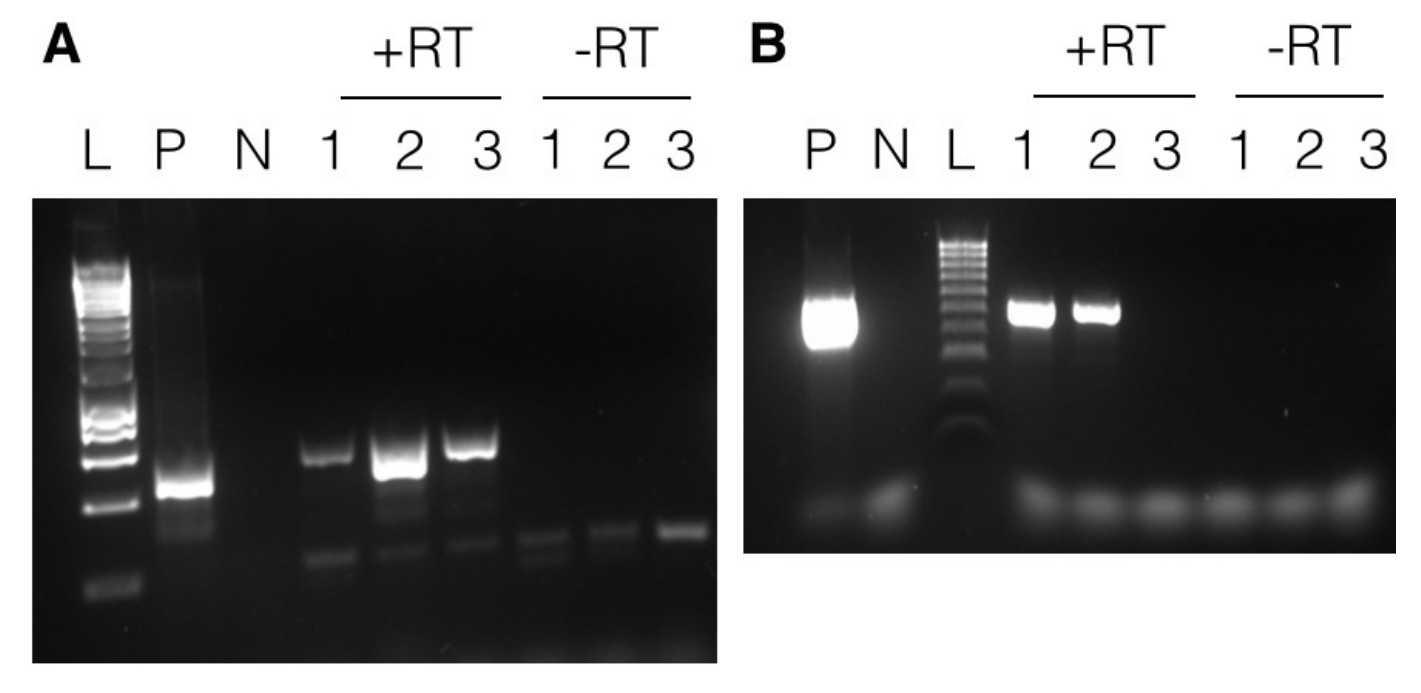

**Figure 4.** Transcription of minicircles. Panel A shows results with pAmpPSBA. RNA was extracted from cells putatively transformed with pAmpPSBA, and RT-PCR performed to amplify a ~ 500 bp region. Lane L, Hyperladder 1 kb (Promega), Lane P, positive control (PCR from plasmid DNA), Lane N, negative control (no template), Lanes 1–3 show products with RNA from three different pAmpPSBA-transformed cell lines (A6.1, A6.2 and A6.3) shown with reverse transcriptase (+RT) and without (-RT). Panel B shows results with pAmpChl. RNA was extracted from cells putatively transformed with pAMPChl, and RT-PCR performed to amplify a 580 bp region. Lane L, Hyperladder 100 bp (Bioline), Lane P, positive control (PCR from plasmid DNA), Lane N, negative control (no template), Lanes 1–3 show products with three different pAmpChl-transformed cell lines (C3.1, C3.2 and C3.3) with reverse transcriptase (+RT) or without (-RT).

DOI: https://doi.org/10.7554/eLife.45292.007

from pAmpChl. The cultures remained alive at the time of writing, 1 year post-transformation. These results show that there was stable transformation of *A. carterae* with pAmpChl.

## Discussion

Here, we present evidence for the first stable transformation of the dinoflagellate chloroplast genome. By making use of the plasmid-like fragmented chloroplast genome and a biolistic system, we have introduced a modified version of an existing sequence as well as a heterologous gene. These genes are transcribed, and produce protein, as shown by immunofluorescence microscopy and the presence of an expected phenotype. Stable transformation was achieved with two separate artificial minicircles, one containing a modified *psbA* gene designed to confer atrazine tolerance and another encoding chloramphenicol resistance, with cultures surviving at least 1 year under selection. The copy number of the artificial minicircle is similar to that of native minicircles.

*A. carterae* cells transformed with the modified *psbA* gene (atrazine tolerance) survived under non-lethal concentrations of atrazine for at least 3 months, retaining the modified gene, indicating that the transformation is stable even under low levels of selection. The results suggest it is important to titrate the concentration of selective agents used. With an atrazine concentration of 2.5 μg $ml^{-1}$ some of the transformed cultures did not survive more than a few days longer than untransformed ones, suggesting that the modified PsbA was at least partially inhibited at that higher atrazine concentration. In addition, it is possible that a background of minicircles containing wild-type *psbA* genes competed with the introduced artificial minicircles for replication or transcription factors, making it difficult for adequate levels of atrazine-insensitive PsbA to be maintained to cope with the higher atrazine concentration. With chloramphenicol concentrations of 30 μg $ml^{-1}$ or above,

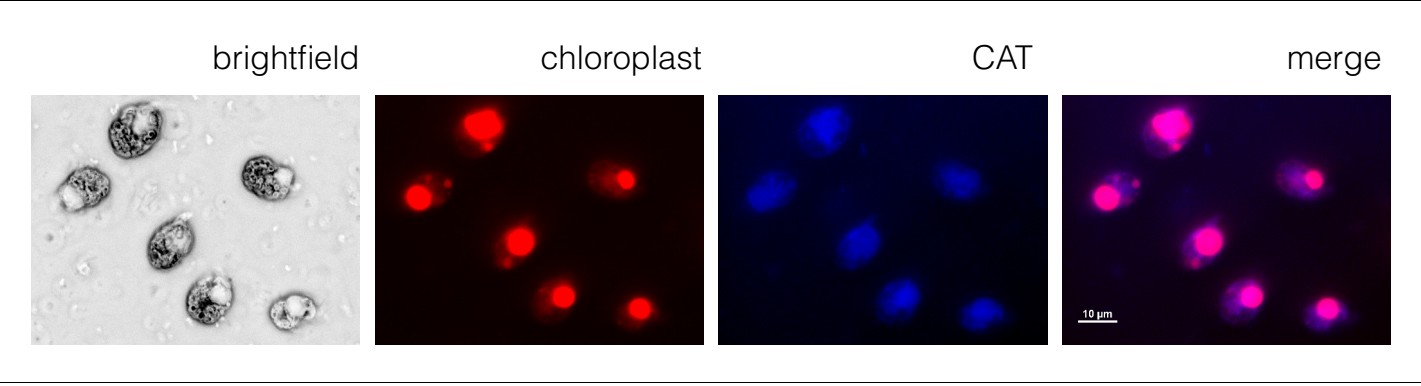

**Figure 5.** A chloroplast localization for chloramphenicol acetyl transferase. Immunofluorescence microscopy using the *A. carterae* pAmpChl line. Cells (brightfield) showed significant autofluorescence in the chloroplast (red). A primary antibody specific for CAT with a secondary Alexa Fluor 405 antibody (blue) showed localization of CAT to the chloroplast (indicated by the overlay image labeled merge).
DOI: https://doi.org/10.7554/eLife.45292.008

the survival of transformed and untransformed strains was similar. However, at 20 μg ml$^{-1}$ or lower the transformed cultures outlasted the untransformed ones, and some were able to survive apparently indefinitely.

There have been multiple previous reports of transformation in dinoflagellates. Although the initial report (using silicon carbide whiskers) described transformation as stable (*ten Lohuis and Miller, 1998*), it has never been successfully reproduced (*Walker et al., 2005*). Transient expression has

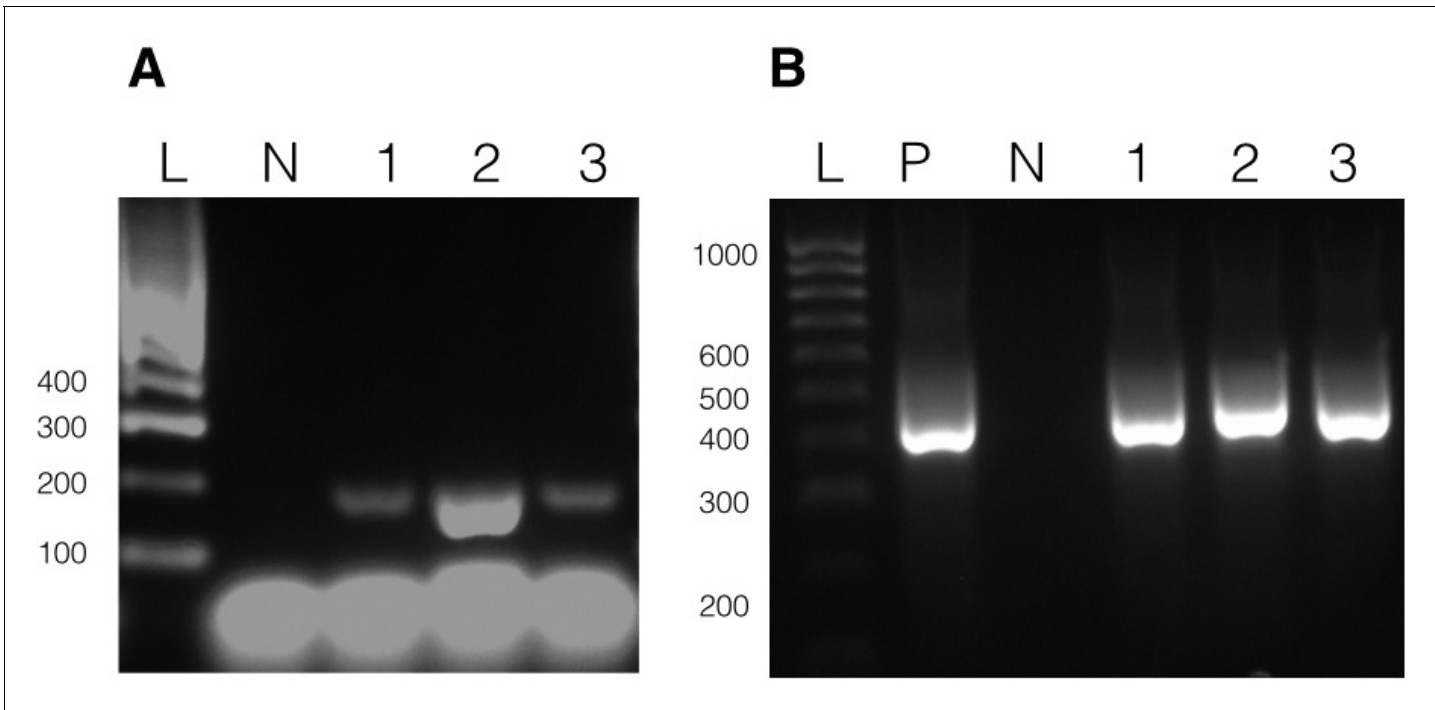

**Figure 6.** Long-term stability of transformation. DNA was isolated from cells putatively transformed with pAmpPSBA or pAmpChl and maintained under selection 3 months (Experiments A9 and C4). Panel A shows PCR to amplify a 200 bp region of pAmpPSBA. Lane L, Hyperladder 100 bp (Bioline) marker, Lane N, untransformed cells, Lane 1, transformed cell line A9.1, Lane 2, transformed cell line A9.2, Lane 3, transformed cell line A9.3. Panel A shows PCR to amplify a 560 bp region of pAmpChl. Lane L, Hyperladder 100 bp (Bioline) marker, Lane P, positive control (pAmpChl), Lane N, untransformed cells, Lane 1, transformed cell line C4.1, Lane 2, transformed cell line C4.2, Lane 3, transformed cell line C4.3.
DOI: https://doi.org/10.7554/eLife.45292.009

been carried out using glass beads (*Ortiz-Matamoros et al., 2015a*). However, we were unable to build on this result to obtain stable transformation, despite numerous attempts (and there have not been any subsequent reports of transformation using either glass beads or *Agrobacterium*). Instead, we found that microparticle bombardment gave stable transformants. Our results – using the same artificial minicircle in all experiments – suggest that the primary reason for failure of the majority of methods to give stable transformation is due to inability of the DNA to enter the cell, rather than inherent dinoflagellate genetics.

The ability to modify the dinoflagellate chloroplast genome will be of enormous value in many areas of dinoflagellate biology. Modification of existing minicircles should allow us to study many other aspects of this highly unusual chloroplast genome, such as the promoter regions of the genes. For example, many chloroplast genes are down-regulated under high temperature stress (*Gierz et al., 2017*). Little is known about how transcription is regulated, or initiated (*Barbrook et al., 2012*), though it is assumed that initiation occurs in the conserved core region of the minicircle (*Barbrook et al., 2001*; *Zhang et al., 2002*). We also do not know how the minicircles are replicated (*Barbrook et al., 2018*), although again it is assumed the core region is important. It will now be possible to mutate this core region to determine which sections are important. The ability to express heterologous proteins will be of great value in studying a wide range of other aspects of dinoflagellate chloroplast biology. The ability to express modified forms of the PsbA protein will be of particular value in studying the role of this protein in the response by dinoflagellates to the disturbances that are believed to precipitate coral bleaching.

# Materials and methods

**Key resources table**

| Reagent type (species) or resource | Designation | Source or reference | Identifiers | Additional information |
|---|---|---|---|---|
| Cell line (*Amphidinium carterae*) | *A. carterae* CCMP1314 | Culture Collection of Marine Phytoplankton | CCMP1314 | |
| Genetic reagent (vector) | pAmpChl | this paper, synthesized by GeneArt | | Full sequence provided in supplemental data. |
| Genetic reagent (vector) | pAmpPSBA | this paper | | Full sequence provided in supplemental data. |
| Antibody | Rabbit anti-Chloramphenicol-acetyl-transferase, polyclonal | Antibodies-Online | Antibodies-Online Cat# ABIN285051, RRID:AB_10781219 | 1:500 in f/2 medium with 5% BSA, 1 hr. |
| Antibody | Goat anti-Rabbit IgG (H + L) Cross-Adsorbed Secondary Antibody, Alexa Fluor 405, polyclonal | ThermoFisher | Thermo Fisher Scientific Cat# A-31556, RRID:AB_221605 | 1:1000 in f/2 medium with 5% BSA, 1 hr. |
| Commercial assay or kit | DNAdelTM Gold Carrier Particles Optimized for Plasmid Delivery | Seashell Technology | Seashell Technology Cat# S550d | |

## Culturing of *Amphidinium carterae*

*A. carterae* CCMP1314 (from the Culture Collection of Marine Phytoplankton) was cultured in f/2 medium on a 16 hr light/8 hr cycle, 18°C at 30µE m$^{-2}$ s$^{-1}$, as described previously (*Barbrook et al., 2006*). Cells were returned to this light regime immediately following all transformation methods.

## Design of artificial minicircles

The pAmpPSBA artificial minicircle (predicted to confer atrazine tolerance) was prepared by PCR amplification of the wild-type *psbA* mincircle with outward facing primers from a point immediately downstream of the proposed poly-U addition site (Genbank AJ250262, fwd primer 1128–1155, rev primer 1127–1106) (*Barbrook et al., 2012*). The linear PCR product was purified and cloned into the pGEM-T Easy plasmid (Promega), which contains the ampicillin resistance marker and a bacterial origin of replication. The point mutations necessary to confer atrazine tolerance were introduced in a further round of PCR with *Pfu* polymerase and the following mutagenic primers, forward primer GTC TTATCTTCCAGTATGCTGGCTTCAACAACTCCCGTTCTC, reverse primer GAGAACGGGAGTTG TTGAAGCCAGCATACTGGAAGATAAGAC. This altered a TCC (Serine) codon to a GGC (Glycine) codon at position 260 of the PsbA protein (numbered as in AJ250262). The PCR products were treated with *Dpn*I to digest any template DNA and then used to transform chemically competent *E. coli* JM109. Ampicillin selection was used to identify colonies containing pAmpPSBA, and plasmids were sequenced. A plasmid map is shown in *Figure 1*. The full vector sequence is given in Supplementary Data.

The pAmpChl vector was synthesized by GeneArt. This vector was based on a pMA vector backbone, and contained the *psbA* minicircle (as above), but with the *psbA* coding region removed and precisely replaced by an *A. carterae* chloroplast codon-optimized *E. coli* chloramphenicol acetyl transferase gene (CAT) (*Barbrook and Howe, 2000*), *Figure 1*. The full vector sequence is given in Supplementary Data.

Both vectors were propagated in *E. coli* under ampicillin selection, and isolated using the Promega Maxiprep plasmid purification protocol prior to transformation into *A. carterae.* The vectors were verified by DNA sequencing before use.

## Glass bead mediated transformation

*A. carterae* cells ($1.3 \times 10^7$) were transformed with pAmpPSBA using glass beads, with or without polyethylene glycol, following the protocol as described by Ortiz-Matamaros et al (*Ortiz-Matamoros et al., 2015a*). A control, where cells were treated with glass beads but without pAmpPSBA, was carried out at the same time. Selection was applied (2 ug/ml atrazine) after 24 hr. Each experiment was carried out in triplicate (i.e. three reactions, three controls).

## Electroporation with Lonza nucleofector

*A. carterae* cells ($1 \times 10^6$) were electroporated using the Lonza Nucleofector, in Lonza media. Settings used were X-100, D-023, L-029 and EH 100. Following electroporation, cells were replaced into f/2 media, and selection applied (2 ug/ml atrazine) after 24 hr. Each experiment was carried out in triplicate (i.e. three reactions, three controls).

## Biolistic transformation of *A. carterae*

Biolistic transformation was carried out using a Biorad Biolistics PDS-1000/He system, Biorad rupture disks, stopping screens and macrocarriers. Preparation of particles carrying DNA was carried out using Seashell Technology's DNAdel gold carrier delivery system and 550 nm diameter gold particles.

*A. carterae* cells were grown to early log growth phase before harvesting prior to transformation. For each transformation, $\sim 2.5 \times 10^7$ cells (as determined by light microscopy utilizing a haemocytometer) were spotted onto the center of a 1% agarose f/2 medium plate and allowed to dry. 0.5 mg of gold particles and 0.5 μg of vector DNA were used for each plate to be transformed. Each plate was shot using the above-mentioned Biorad Biolistics PDS-1000/He system and rupture disks of either 1100 PSI, 1350 PSI or 1550 pounds per square inch (p.s.i.) (see *Table 1* for details).

Cells were immediately resuspended in 30–50 ml fresh f/2 medium and allowed to recover before the addition of the selective agent. Cells shot using the pAmpPSBA artificial minicircle were allowed 16–24 hr to recover. Cells shot using the pAmpChl artificial minicircle were allowed 72 hr to recover. Cells were maintained in liquid culture as they do not grow on solid medium. Medium was replaced every 4 weeks (atrazine) or 2 weeks (chloramphenicol). Cells were subcultured (two fold dilution) every 8 weeks (atrazine) or 4 weeks (chloramphenicol). A step-by step protocol is described in

'Biolistic Transformation of Amphidinium' (https://www.protocols.io/view/biolistic-transformation-of-amphidnium-hnmb5c6).

Culture survival was assessed by microscopy. A spot of 50 µl was placed onto each of three microscope slides for each culture. After covering with standard coverslips, the entire volume for each was examined using a light microscope at x100 magnification. Cells were assessed as living if they showed more than simple Brownian motion. In general, 'dead' cells appeared to disintegrate shortly after movement ceased. If no living cells were found 3 days in a row, the culture was recorded as dead on day 3.

## Extraction of DNA and RNA from *A. carterae*

Total RNA was isolated from *A. carterae* using the Trizol - chloroform method. Purification was carried out using the RNA clean-up with on-column DNase protocol of the Qiagen RNeasy kit as described (*Rio et al., 2010*) except that isopropanol precipitation was carried out overnight, rather than for 10 min.

DNA was released from cells prior to PCR by resuspending $5 \times 10^4$ to $10^7$ cells (depending on the number available) in 50 µl dH2O with ~10–20 acid-washed 500 µm glass beads and vortexing for 10 min.

## RT-PCR and PCR

First strand synthesis of the RNA was performed using Invitrogen Superscript IV using the manufacturer's protocol and either random hexamer primers or a gene-specific primer. Negative controls lacking reverse transcriptase were performed by the same method but replacing the reverse transcriptase enzyme with dH$_2$O. PCR was carried out using Promega GoTaq polymerase according to the manufacturer's instructions, and annealing temperature, extension time and MgCl$_2$ concentration were varied as appropriate.

## Cloning and sequencing of PCR products

PCR products were separated by 1–1.5% agarose gel electrophoresis and visualized by staining with GelRed. PCR products were purified from excised gel pieces using the MinElute gel extraction kit (Qiagen). Some PCR products were directly sequenced after gel extraction whilst others were ligated into the pGEM-T Easy plasmid vector (Promega), following the manufacturer's instructions. The ligation mix was used to transform chemically competent *Escherichia coli* TG1, followed by overnight growth on 1.5% LB agar containing ampicillin at 100 µg/ml. Individual colonies were picked and grown overnight in LB containing ampicillin at 100 µg/ml. Plasmids were extracted from resulting cultures using the QIAprep Spin Miniprep Kit (Qiagen). All sequencing was carried out using an Applied Biosystems 3130XL DNA Analyser in the Department of Biochemistry, University of Cambridge sequencing facility.

## Immunofluorescence microscopy

Five $\times 10^5$ cells were fixed in 1% paraformaldehyde in f/2 medium for 5 min. The reaction was quenched by the addition of glycine to a final concentration of 0.125 M for a further 5 min. Both steps were carried out with constant agitation. Cells were washed in f/2 medium three times for 5 min each, and then permeabilised by addition of 0.2% Triton-X-100 for 15 min, again with constant agitation, and washed three further times in f/2 medium. Blocking was carried out in 5% BSA in f/2 for 30 min under constant agitation.

Cells were incubated with a rabbit anti-chloramphenicol acetyl transferase as the primary antibody (Antibodies online, ABIN285051) at a final concentration of 1:500 in f/2% and 5% BSA for 1 hr under constant agitation, and washed three times in f/2 medium for 5 min. Cells were then incubated with an anti-rabbit Alexa Fluor 405 secondary antibody (raised in goat; ThermoFisher) at a final concentration of 1:1000 in f/2% and 5% BSA for 1 hr under constant agitation, washed in f/2 medium three times for 5 min, and mounted on VWR-polysine-coated slides using Vectashield mounting medium. Cells were visualized using a Nikon C2 confocal microscope.

## Acknowledgements

This research is funded by the Gordon and Betty Moore Foundation through Grant GBMF4976.01 to CJH, RFW, SP and MA. JEC was supported by the King Abdullah University of Science and Technology (KAUST) Office of Sponsored Research (OSR) under Award No. URF/1/2216-01-01. We thank Nathan Parker for technical assistance.

## Additional information

### Funding

| Funder | Grant reference number | Author |
| --- | --- | --- |
| Gordon and Betty Moore Foundation | GBMF4976.01 | Isabel C Nimmo<br>Adrian C Barbrook<br>Imen Lassadi<br>Manuel Aranda<br>Saul Purton<br>Ross F Waller<br>R Ellen R Nisbet<br>Christopher J Howe |
| King Abdullah University of Science and Technology | URF/1/2216/01-01 | Jit Ern Chen |

The funders had no role in study design, data collection and interpretation, or the decision to submit the work for publication.

### Author contributions

Isabel C Nimmo, Formal analysis, Investigation, Methodology, Writing—review and editing; Adrian C Barbrook, Conceptualization, Investigation, Methodology, Project administration, Writing—review and editing; Imen Lassadi, Formal analysis, Methodology; Jit Ern Chen, Methodology; Katrin Geisler, Resources; Alison G Smith, Conceptualization, Resources, Funding acquisition, Writing—review and editing; Manuel Aranda, Saul Purton, Conceptualization, Funding acquisition; Ross F Waller, Conceptualization, Funding acquisition, Project administration; R Ellen R Nisbet, Conceptualization, Supervision, Funding acquisition, Investigation, Methodology, Writing—original draft, Project administration, Writing—review and editing; Christopher J Howe, Conceptualization, Supervision, Funding acquisition, Methodology, Project administration, Writing—review and editing

### Author ORCIDs

Isabel C Nimmo https://orcid.org/0000-0002-2469-0611
Jit Ern Chen http://orcid.org/0000-0003-4779-7275
Manuel Aranda http://orcid.org/0000-0001-6673-016X
Saul Purton https://orcid.org/0000-0002-9342-1773
Ross F Waller https://orcid.org/0000-0001-6961-9344
R Ellen R Nisbet https://orcid.org/0000-0003-4487-196X
Christopher J Howe https://orcid.org/0000-0002-6975-8640

### Decision letter and Author response

Decision letter https://doi.org/10.7554/eLife.45292.013
Author response https://doi.org/10.7554/eLife.45292.014

## Additional files

### Supplementary files

• Supplementary file 1. Primer sequences and vector sequences.
DOI: https://doi.org/10.7554/eLife.45292.010

• Transparent reporting form
DOI: https://doi.org/10.7554/eLife.45292.011

**Data availability**

Sequencing data are included in the manuscript.

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
