## [Decision Letter]

Thank you for submitting your article "Genetic transformation of the dinoflagellate chloroplast" for consideration by *eLife*. Your article has been reviewed by two peer reviewers, and the evaluation has been overseen by a Reviewing Editor and Christian Hardtke as the Senior Editor. The reviewers have opted to remain anonymous.

The reviewers have discussed the reviews with one another and the editors, and we have drafted this decision to help you prepare a revised submission.

The reviewers and editors agree that your paper represents an important technical advance in dinoflagellate biology that merits publication as a "Tools and Resources" manuscript (although all would have liked to see some additional biological insight from your study). There are however a few essential revisions we would like to ask you to perform before your work can be accepted for publication. Specifically:

- Although you demonstrate transformation of the CAT marker, you have not yet definitively shown that the CAT plasmid was located in the plastid. Additional experimental data that proof this point should be provided.

- Since there is just one plastid per cell with many copies of each minicircle (as shown by your own previous work), one would expect that each surviving plastid must have a mixed population of original and transformed circles. To fully characterize the successful transformants, it therefore seems important that you determine the copy number of wild-type and introduced plasmids and/or their ratio.

Besides these essential revisions, please find additional suggestions for improvement in the individual reviews attached below.

*Reviewer #1:*

The authors are to be commended for embarking on a difficult project where there is a (largely unpublished) history of failed attempts. This paper reports significant preliminary progress toward the goal of stable chloroplast transformation.

On the whole, this paper has shown that stable transformation of the Amphidinium carterae plastid with a mutated psbA gene is possible. About the transformation of the CAT marker, the cells have been transformed although it was not definitively shown that the CAT plasmid was located in the plastid. And the authors missed the opportunity to query those cells about their normal psbA-an interesting evolutionary story waiting for some more experiments.

There are some points to be addressed:

1) It was not explicitly stated exactly what the culture conditions were for the cells that had just been subject to transformation --were they put right back in the same light intensity as the parent cultures? That seems a bit rough.

2) Table 2 – it might have been informative if a sample of the mock-transformed cells had been allowed to grow without selection, to provide a growth-rate comparison for the transformants that survived selection, e.g. in experiment A5, what was the cell density of the transformed culture on the day of harvest, compared with the cell density of the mock culture growing "normally"?

3) In experiments A6 and A7, both mock-transformed and plasmid-transformed cells were harvested while the cultures were still alive, but what was the cell density of each?

About A9: If subculturing was at "8-week intervals" that means the cells that survived 3 months had only been transferred once!

4) About the PCR results in Figure 2A. the band for 5.2 is so faint it's almost invisible. Perhaps add a comment to that effect.

5) pAmpChl plasmid doesn't carry a psbA gene, so in order to survive the cells would have to maintain a population of their original minicircles with the normal psbA gene. Since there is just one plastid per cell with many copies of each minicircle (as shown by some of these authors some years ago) this means that each surviving plastid must have a mixed population of original and transformed circles. So it would be important to know what that ratio is, and PCR should have been done to detect the native psbA. Again, knowing whether the cells were dividing or just surviving is important.

*Reviewer #2:*

This manuscript is well written and provides an important step forward in the field of algal biotechnology. The ultimate stated goal of improving coral reef resilience by "designing" more robust algal symbiont is less relevant at this juncture. The methods are clearly described and rely on standard approaches.

However, the lack of any biologically relevant data in terms of engineering traits beyond antibiotic resistance makes this paper less interesting than it could have been. The poor state of knowledge about dinoflagellate minicircle gene expression regulation makes this an issue for this and future work. Therefore, some mutagenesis work using the pAmpPSBA plasmid focusing on the core region implicated in transcription initiation would have been valuable.

The other issue that perhaps only needs some explanation is evidence about where the minicircles end up in the cell. This work is cited as chloroplast genome transformation but are the authors certain that minicircles also do not localize to the nucleus or cytosol? In plants, it is clear that organelle genomes are organelle-bound, but I am not sure this is the case for dinoflagellate minicircles. In situ hybridization would address this concern.

Recent work on Porphyridium transformation showed that the plasmids were episomally localized in the nucleus.

Did the authors study minicircle (wt and introduced) copy number during the experiment? Could this explain the low growth rate or other aspects of the transformation?

---

## [Author Response]

The reviewers and editors agree that your paper represents an important technical advance in dinoflagellate biology that merits publication as a "Tools and Resources" manuscript (although all would have liked to see some additional biological insight from your study). There are however a few essential revisions we would like to ask you to perform before your work can be accepted for publication. Specifically:- Although you demonstrate transformation of the CAT marker, you have not yet definitively shown that the CAT plasmid was located in the plastid. Additional experimental data that proof this point should be provided.

IFAs show that expressed proteins from the construct are clearly located in the plastid, as discussed below, and shown in Figure 5.

- Since there is just one plastid per cell with many copies of each minicircle (as shown by your own previous work), one would expect that each surviving plastid must have a mixed population of original and transformed circles. To fully characterize the successful transformants, it therefore seems important that you determine the copy number of wild-type and introduced plasmids and/or their ratio.

The copy number of individual minicircles varies considerably across the life cycle, from 1 or 2 to several hundred (see our previous paper, Koumandou and Howe, 2007). Therefore, determining the copy number doesn’t really give much useful information, aside from giving an indication of the growth stage of the culture (log/lag/stationary). What is relevant however is how the copy number compares to that of the native psbA minicircle that we have hijacked to create the artificial minicircle backbone. We therefore carried out a semi-quantitative PCR reaction with primers immediately adjacent to the gene (either psbA or CAT), in a shared region of the minicircle. This revealed that the larger band (psbA) was twice the intensity of the lower band (CAT). Although they differ slightly in size, there is no reason to suspect a major difference in PCR efficiency. As the CAT gene is half the size of psbA, this would therefore indicate that about half the minicircles with a psbA backbone contain psbA, and half contain CAT. This data is included in Figure 3.

Besides these essential revisions, please find additional suggestions for improvement in the individual reviews attached below.

Reviewer #1:

The authors are to be commended for embarking on a difficult project where there is a (largely unpublished) history of failed attempts. This paper reports significant preliminary progress toward the goal of stable chloroplast transformation.On the whole, this paper has shown that stable transformation of the Amphidinium carterae plastid with a mutated psbA gene is possible. About the transformation of the CAT marker, the cells have been transformed although it was not definitively shown that the CAT plasmid was located in the plastid. And the authors missed the opportunity to query those cells about their normal psbA-an interesting evolutionary story waiting for some more experiments.There are some points to be addressed:1) It was not explicitly stated exactly what the culture conditions were for the cells that had just been subject to transformation --were they put right back in the same light intensity as the parent cultures? That seems a bit rough.

Yes, they were put straight back into the same conditions as before, although selection was not immediately applied, to give the chance for the cells to recover. We have included a statement in the Materials and methods.

2) Table 2 – it might have been informative if a sample of the mock-transformed cells had been allowed to grow without selection, to provide a growth-rate comparison for the transformants that survived selection, e.g. in experiment A5, what was the cell density of the transformed culture on the day of harvest, compared with the cell density of the mock culture growing "normally"?

The transformation protocol is highly variable in its efficiency, primarily due to the difficulties of biolistics. This requires cells to be on an agar plate. Amphidinium will only grow in liquid culture. Cells therefore need to be plated out, taken for biolistic transformation (in a vacuum) and then scraped off the agar plate and resuspended into liquid culture. There are consequently a large number of variables that prevent meaningful comparison of the growth and survival of the cultures. The survival rate is therefore very variable, based on the time not in solution (the first transformed line always does better than the last; there is an upper limit of 12 transformations that can be carried out at any one time). Thus, measuring the cell density is not a very accurate measure of anything. However, longer-term, the cell growth rate can be compared. Atrazine-selected lines do grow significantly more slowly than wild type, while chloramphenicol-selected lines grow a bit more slowly than wild type.

3) In experiments A6 and A7, both mock-transformed and plasmid-transformed cells were harvested while the cultures were still alive, but what was the cell density of each?

About 50-100,000 cells/ml. Though this is affected by the variables described above.

About A9: If subculturing was at "8-week intervals" that means the cells that survived 3 months had only been transferred once!

One of the issues with the pAmpPSBA lines is that they grew extremely slowly. The cells were transformed, allowed to grow, subcultured and allowed to grow. They were harvested just before the second subculturing was due. Cell growth was monitored by microscopy, with cell counts taken at 2-week intervals, and cell density was increasing.

4) About the PCR results in Figure 2A. the band for 5.2 is so faint it's almost invisible. Perhaps add a comment to that effect.

Done.

5) pAmpChl plasmid doesn't carry a psbA gene, so in order to survive the cells would have to maintain a population of their original minicircles with the normal psbA gene. Since there is just one plastid per cell with many copies of each minicircle (as shown by some of these authors some years ago) this means that each surviving plastid must have a mixed population of original and transformed circles. So it would be important to know what that ratio is, and PCR should have been done to detect the native psbA. Again, knowing whether the cells were dividing or just surviving is important.

We have carried out this PCR, as suggested, and include a new figure. The ratio of psbA minicircle to pAmpChl minicircle is about 1:1.

Reviewer #2:

This manuscript is well written and provides an important step forward in the field of algal biotechnology. The ultimate stated goal of improving coral reef resilience by "designing" more robust algal symbiont is less relevant at this juncture. The methods are clearly described and rely on standard approaches.However, the lack of any biologically relevant data in terms of engineering traits beyond antibiotic resistance makes this paper less interesting than it could have been. The poor state of knowledge about dinoflagellate minicircle gene expression regulation makes this an issue for this and future work. Therefore, some mutagenesis work using the pAmpPSBA plasmid focusing on the core region implicated in transcription initiation would have been valuable.

We agree, and this is the next step. However, we feel it important to publish the technique so that others can make use of it, rather than waiting for these experiments to conclude. The experiments are in progress, and we will publish them once they are completed.

The other issue that perhaps only needs some explanation is evidence about where the minicircles end up in the cell. This work is cited as chloroplast genome transformation but are the authors certain that minicircles also do not localize to the nucleus or cytosol? In plants, it is clear that organelle genomes are organelle-bound, but I am not sure this is the case for dinoflagellate minicircles. In situ hybridization would address this concern.Recent work on Porphyridium transformation showed that the plasmids were episomally localized in the nucleus.

Dinoflagellate minicircles are indeed the chloroplast, as shown by Owari et al., 2013, in Amphidinium using in situ hybridization. This includes the psbA minicircle, which is the one we selected for modification. We therefore carried out immunofluorescence looking for the location of chloramphenicol acetyl transferase protein using an antibody. It co-localizes to regions of the cell with autofluorescence and thus is in the chloroplast.

Did the authors study minicircle (wt and introduced) copy number during the experiment? Could this explain the low growth rate or other aspects of the transformation?

Yes, it is about 50% wild type psbA minicircle, 50% pAmpChl minicircle (see above). We think that the problem with growth rate for the pAmpPSBA minicircle is due to the mutation that we introduced. There are numerous psbA mutations which give rise to atrazine resistance in plants. The mutation we selected for use in the construct is the one that gives the highest level of resistance (Oettmeier et al. 1999, Cell Mol Life Sci 55:1255-1277). However, other research has shown that it can have consequences for photosynthesis (Erickson et al. 1989, Plant Cell 1:361-371). In many plant and algal species, this does not appear to be significant for viability, but it does appear to be more significant for Amphidinium.